# Effect of Thermo-Physical Properties of the Tool Materials on the Electro-Discharge Machining Performance of Ti-6Al-4V and SS316 Work Piece Materials

Sunita Sethy [1], Rajesh Kumar Behera [2,*], Jõao Paulo Davim [3] and Jaydev Rana [1,*]

1 Department of Mechanical Engineering, Veer Surendra Sai University of Technology, Burla, Sambalpur 768018, India
2 Department of Mechanical Engineering, Orissa Engineering College, Bhubaneswar 752050, India
3 Department of Mechanical Engineering, University of Aveiro, Campus Santiago, 3810-193 Aveiro, Portugal
* Correspondence: rajesh_k_behera@yahoo.co.in (R.K.B.); jranauce@yahoo.co.in (J.R.)

**Abstract:** Electro-discharge machining (EDM) is a useful non-conventional machining operation frequently applied to make different complex geometries in any conducting material. The objectives of the present paper are to study the effect of a variation of thermo-physical properties (TPP) of three different tool materials on EDM performances. The different performances compared in this paper are: material removal rate (MRR), tool-wear rate (TWR), surface roughness (SR), radial overcut (ROC), surface-crack density (SCD) and surface hardness. Two of the most widely used work piece materials, such as corrosion-resistant austenitic stainless steel (SS316) and high strength corrosion-resistance titanium alloy (Ti-6Al-4V), are machined with the help of three different tools by varying input current and maintaining constant pulse-on time, pulse-off time and flushing pressure. Microstructural studies of the tool tip surface after machining have also been carried out. It is found that among these three tool materials, the copper tool showed the best machining performance with respect to material removal rate, radial overcut, surface finish and surface-crack density. This work will help industry personnel to choose a suitable tool for a specific work piece material.

**Keywords:** electro-discharge machining; material removal rate; tool-wear rate; surface roughness; radial overcut; surface-crack density

## 1. Introduction

Electro-discharge machining (EDM) is an electro-thermal process that uses pulsed electrical energy. This pulsed electrical energy results in sparks and strikes the work piece surface. The approximate temperature where these sparks hit the work piece surface is of the order of 6000–12,000 °C [1]. As a consequence, a small part of the work piece material is evaporated and melted. As soon as the electrical energy is discontinued, the majority of the parts of molten materials and all evaporated materials are ejected out from the work piece. It leaves craters on the work piece surface. This process takes place within a few micro-seconds.

Many previous researchers have changed the tool materials and studied the machining performances with variation of the tool. Bhaumik and Maity investigated the effect of the EDM process parameter as well as the different types of electrode on the surface integrity and dimensional accuracy of Ti-5Al-2.5Sn titanium alloy after machining in EDM. They had used copper, brass and zinc electrodes. According to them, the copper electrode offered a good surface finish and less radial overcut than brass and zinc electrodes. Moreover, they also found that a very thin and uniform recast layer with higher surface-crack density had been observed in the case of copper electrode [2]. Bhaumik and Maityalso studied the effect of different tool materials such as copper, brass and zinc on the machining performance of Ti-5Al-2.5Sn. In that paper, MRR and TWR had been considered under

machining performances. They came to know that higher MRR was obtained by brass and zinc compared to the Cu electrode. However, the Cu electrode was showing a lesser tool-wear rate, followed by brass and zinc [3]. Rahul et al. investigated the effect of different tool electrodes such as tungsten, Cu and cryogenically treated Cu on the machinability of Ti-6Al-4V material in EDM. They came to know that cryogenically treated Cu was showing better machinability [4]. Ahmed et al. studied the machinability of titanium alloy using Cu, Al, brass and graphite electrodes. They came to know that the graphite electrode had high MRR with low surface roughness by initially employing negative tool polarity for rough machining and then positive tool polarity for fine machining [5]. Sen and Mondal investigated the effect of different tool electrodes such as copper, graphite and brass on the performance of EDM while machining mild steel with an IS2062 grade. They had studied MRR, TWR and SR by varying the peak current. They claimed that graphite was showing the highest MRR, whereas brass was showing good surface finish [6]. Kishawy et al. performed a sustainability assessment while machining Ti-6Al-4V with nano-additive-based minimum-quantity lubrications. The sustainability aspect included the environmental impact, management of waste, and safety and health issues of the operator, in order to validate the effectiveness of sustainability results, and a comparison between optimal and predicted responses was conducted and had good agreement [7]. Chaudhari et al. performed multi-response optimization of process parameters in the wear EDM process while machining a super-elastic Nitinol shaped-memory alloy (Ni55.8Ti). A multi-objective heat-transfer search algorithm was executed for generating 2D and 3D Pareto optimal points indicating the non-dominant feasible solution. The optimized parameters were found to machine the alloy appropriately, keeping intact the shape memory effect [8]. Walia et al. had studied the distortion in a tool set during the machining of EL31 tool steel. The change in out-of-roundness of the tool tip had been found to vary from 5.65 to 37.8 micrometers. They claimed that the input current, the pulse-off time and pulse-on time were most significant in changing the out-of-roundness value during machining [9]. Philip et al. had compared the EDM performances such as MRR, TWR, SR, microstructure and surface integrity while machining Ti-6Al-4V and other work pieces in simple EDM and powder-mixed EDM [10]. Doreswamy et al. had investigated the machinability of silicon particle-reinforced Al6061 composite by a wire-EDM process. They investigated the effect of current, wire speed, pulse-on time, pulse-off time and voltage on MRR [11]. Roy and Dutta had optimized the MRR, TWR, and tool overcut and reported the optimum levels of input parameters. They claimed that the discharge current was showing the highest contribution among pulse-on time, duty cycle, gap voltage and discharge current [12]. Kumar et al. studied the machinability aspect, such as the surface quality of titanium-based human implant material, using wire-EDM process. They had varied pulse-on time, pulse-off time and voltage during the experimentation [13]. Swiercz et al. optimized surface roughness, MRR and white layer thickness while machining tool steel in EDM using the desirability function. They had also evaluated the surface and sub-surface integrity using an optical microscope and scanning profilometer [14]. Tiwary et al. mixed different conducting powders such as cobalt, nickel and copper in deionized water with changing concentration and optimized surface roughness, tool-wear rate, MRR, taper and overcut using the principal component analysis method [15]. Qudeiri et al. improved the machining performance such as MRR, surface quality and TWR of different grades of stainless steel in EDM by suitably selecting the input parameters and work piece materials [16]. Nair et al. carried out a machinability study on Ti-6Al-4V material in EDM. They studied the effect of discharge voltage, current and discharge time on SR, micro-cracks, white layer thickness and blowholes on the machined specimen. They claimed that with increasing current and discharge time, white layer thickness and MRR were improved [17]. Ahmed et al. carried out EDM of Ti-6Al-4V with two alternate polarities and selected appropriate tool materials among Al, Cu and brass on the basis of minimum TWR and overcut [18]. Sharma et al. conducted process optimization while machining WC. They optimized SR and micro-hardness using grey relational analysis [19]. Jadam and Datta

had investigated the machinability of Ti-5Al-2.5Sn in EDM. They claimed that the EDM improved the SH by about three times due to formation of titanium carbide [20]. Illani and Khoshnevisan found that there is an improvement for MRR, TWR and SR by 33%, 31% and 77%, respectively [21]. Naik et al. optimized the surface quality and hole quality while machining Al-22%SiC MMC in EDM. They claimed that discharge current was significant in affecting the above performance characteristics [22]. Bui et al. performed anti-bacterial coating on the surface of Ti-6Al-4V. They used silver nano-powder in the dielectric medium. They claimed that due to the suspension of silver particles in the dielectric, the machining performance was improved [23]. Singh et al. used the EPSDE technique to optimize the TWR and MRR while machining Ti-6Al-4V material in EDM [24]. Devarasiddappa et al. investigated experimentally the machining performance of Ti-6Al-4V alloy using the wire-cut EDM process. They used the TLBO algorithm for the optimization of SR and MRR [25]. Abdudeen et al. reported the recent advances in the powder-mixed dielectric of the EDM process. They claimed that due to the mixing of powder in the dielectric, the SR and MRR were improved with the reduction of TWR [26]. Kumar et al. had modified the surface of the work piece during the EDM process. They had produced an electrode through the powder metallurgy process and tested the EDM performance during machining. They came to know that this new technique enhanced the micro-hardness, surface finish, wear-resistance and corrosion behavior of the material by surface modification [27]. Chandrashekarappa et al. had carried out a comparative study on EDM taking HcHcrD2 steel by considering different electrode materials such as copper, graphite and brass. They claimed that the graphite electrode was showing the best performance with respect to MRR, TWR and SR [28]. Sahu and Mahapatra had studied the performance of electrodes prepared through the laser sintering process in EDM while machining titanium. The performance of the newly prepared electrode through the laser sintering process using a metal matrix composite of AlSiMg with copper and graphite electrodes was observed and they claimed that the micro-hardness of the machined surface was increased due to formation of titanium carbide on the machined surface, and the surface produced when machined with the RP tool electrode exhibited superior surface characteristics compared to copper and graphite [29].

It is observed from the previous literature that the previous researchers had tried to study the performance of different tools and work piece combinations. They had reported the performance in a different way but the basis of the difference in performance was not reported. In this present research work, three different tool electrodes, such as Al, brass and copper, have been chosen, with a variation of thermo-physical properties (TPP) such as melting point, thermal conductivity and thermal diffusivity. On the basis of these properties, the performance variations of two commonly used high-strength work piece materials such as Ti-6Al-4V and SS316 have been carried out. A titanium alloy such as Ti-6Al-4V is widely used in aerospace, automobile industries and medical applications because of its high strength-to-weight ratio, excellent corrosion- and wear-resistance, high fatigue strength and very good bio-compatibility properties. Stainless steel 316 (i.e., SS316) is mostly used in aerospace structures. Due to the above versatile usefulness of these two materials, it has been decided to perform experiments on these two work piece materials with a variation of tool electrodes. These combinations of tools and work pieces have not been tested by previous researchers. In order to simplify the experimental work, the current study is varied at four different steps, keeping the other input parameters as constant because current is the most vital input parameter affecting the EDM performance. In the present research work, six output parameters such as MRR, SR, TWR, ROC, SCD, SH have been studied with variations in different tool electrodes. These many parameters have not been studied earlier with variations in tool electrodes.

## 2. Experimentation

### 2.1. Work Piece Materials

In the present investigation, two work piece materials such as Ti-6Al-4V and SS316 were used for machining. The chemical compositions of different work piece samples are presented in Tables 1 and 2, with corresponding EDS analysis (shown in Figures 1 and 2).

**Table 1.** Chemical composition of Ti-6Al-4V.

| Element | Weight% | Atomic% |
|---------|---------|---------|
| Al | 10.35 | 17.04 |
| Ti | 87.13 | 80.77 |
| V | 2.52 | 2.19 |
| Totals | 100.00 | |

**Table 2.** Chemical composition of SS316.

| Element | Weight% | Atomic% |
|---------|---------|---------|
| Si | 1.05 | 2.04 |
| S | 0.74 | 1.27 |
| Cr | 18.82 | 19.82 |
| Fe | 68.32 | 67.00 |
| Ni | 9.79 | 9.14 |
| Mo | 1.28 | 0.73 |
| Totals | 100.00 | |

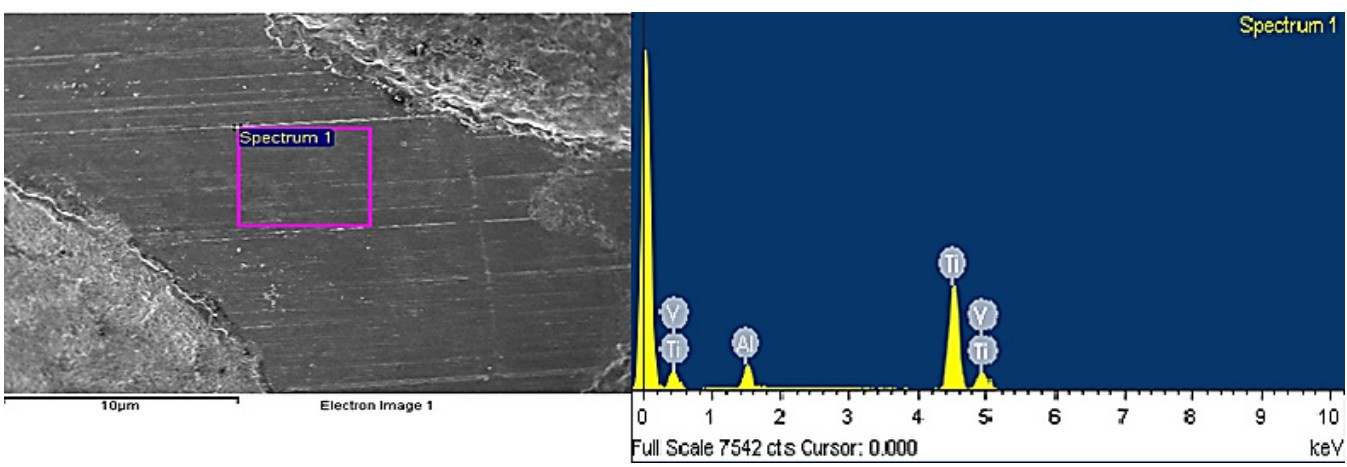

**Figure 1.** EDS spectra of Ti-6Al-4V.

The micro-hardness value of different work piece materials prior to machining has been determined with the help of a micro-hardness tester, and the values are 410.05 HV for Ti-6Al-4V material and 328.9 HV for SS316 material. The dimensions and the TPP of two work pieces have been presented in Table 3.

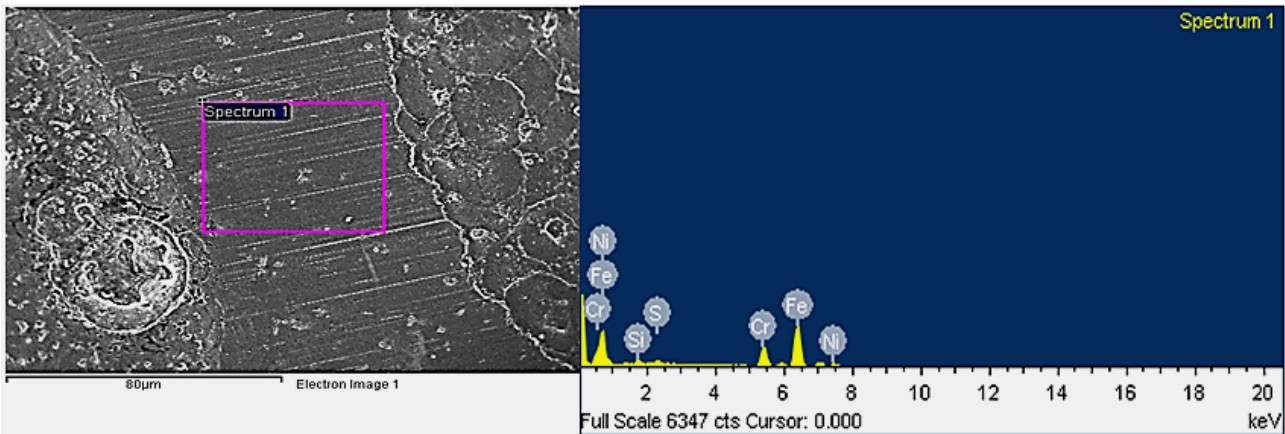

**Figure 2.** EDS spectra of SS316.

**Table 3.** Dimension and the thermo-physical properties of different work pieces.

| Sl. No. | Type of Work Piece | Dimension (mm) | TPP | | |
|---|---|---|---|---|---|
| | | | Thermal Conductivity (W/mK) | Thermal Diffusivity ($m^2/s$ $10^{-7}$) | Melting Point (K) |
| 1. | Ti-6Al-4V | $120 \times 20 \times 5$ | 7.10 | 28 | 1299.15 |
| 2. | SS316 | $120 \times 20 \times 5$ | 13.0 | 2.68 | 1673.15 |

It is observed from the thermal conductivity values of Ti-6Al-4V and SS316 that the thermal conductivity value of SS316 is about two times that of Ti-6Al-4V and the thermal diffusivity value of SS316 is about one tenth of that of the Ti-6Al-4V material.

### 2.2. Tool Materials

Aluminium, copper and brass tools were used in the present experiment and the faces of the tools were machined to achieve uniform surfaces. Each tool had a 10 mm diameter with 80 mm length. The different physical-properties of the tool electrodes are presented in Table 4.

**Table 4.** The physical properties of Aluminium, Brass and Copper tool materials.

| Sl. No. | Physical Properties | Tool Materials | | |
|---|---|---|---|---|
| | | Aluminium (Al) | Brass | Copper (Cu) |
| 1 | Melting Point | 933.15 K | 1203.15 K | 1357.77K |
| 2 | Density | 2.7 g/cm$^3$ | 8.73 g/cm$^3$ | 8.96 g/cm$^3$ |
| 3 | Heat of Fusion | 10.7 kJ/mol | 168 kJ/mol | 13.26 kJ/mol |
| 4 | Thermal conductivity | 237 W/mK | 120 W/mK | 386 W/mK |
| 5 | Thermal diffusivity | $9.7 \times 10^{-5}$ m$^2$/s | $3.75 \times 10^{-5}$ m$^2$/s | $11 \times 10^{-5}$m$^2$/s |
| 6 | Electrical conductivity | $62.1 \times 10^6$ Siemens/m | $36.9 \times 10^6$ Siemens/m | $15.9 \times 10^6$ Siemens/m |

### 2.3. Plan of Experiment

In the present experimental work, three tool materials such as Al, brass and Cu have been used. Each tool will be used to machine two different work piece materials such as Ti-6Al-4V and SS316. In this present experimental set up, three input parameters such as current, pulse-on time and pulse-off time will be considered. In order to simplify the experiment, only current is varied at four steps, such as 5, 10, 15, 20, maintaining constant values of pulse-on time and pulse-off time (i.e., 1000 μs and 2000 μs, respectively). It

may also be noted that the dielectric flushing pressure and tool feed rate are not changed during the experiments. Since current is the most influencing parameter, only current is varied in the present experimentation. The different sets of input parameters are presented in Table 5.

**Table 5.** Different sets of input parameters.

| Experiment No. | Input Parameters | | |
|:---:|:---:|:---:|:---:|
| | I (A) | $T_{on}$ (µs) | $T_{off}$ (µs) |
| 1 | 5 | 1000 | 2000 |
| 2 | 10 | 1000 | 2000 |
| 3 | 15 | 1000 | 2000 |
| 4 | 20 | 1000 | 2000 |

*2.4. Experimental Set up and Procedure*

The experiment has been carried out with the help of a die sinking EDM machine. The detailed experimental methodology as well as the equipment proposed is schematically shown in Figure 3.

In order to perform the experiments, at first, the machine was readied. Then the weights of the tools and work pieces were measured with the help of a micro-balance. Subsequently, the work piece and the tool were fitted in the fixture and the tool holder, respectively. After setting appropriate input parameters in the control panel, machining was carried out for a particular period of time and the machining time was noted with the help of a stopwatch. After machining was over, the work piece and tools were taken out of the machine and the final weights of tool and work piece were measured with the help of a micro-balance. The initial weight and the final weight of the work piece and the tool were recorded. In this way, all experiments were performed for all sets of input parameters and for all sets of tool and work piece combinations. After machining was over, both the work pieces were taken out, and the photographs showing the machined area on the work piece of different experiments are shown in Figure 4.

*2.5. Experimental Results*

The different output parameters are determined as stated in the following:

1. The MRR is calculated by using Equation (1) as stated in the following:

$$\text{Material removal rate} = \frac{\text{Initial Wt. of workpiece} - \text{Final Wt. of workpiece}}{\text{Time of machining}} \quad (1)$$

Putting the values of initial weight, final weight and time of machining the MRR can be calculated.

2. The TWR is calculated by using Equation (2) as stated in the following:

$$\text{Tool wear rate} = \frac{\text{Wt. of tool before machining} - \text{Wt. of tool after machining}}{\text{Time of machining}} \quad (2)$$

By inputting the values of initial weight of tool, final weight of tool and time of machining, the TWR can be calculated.

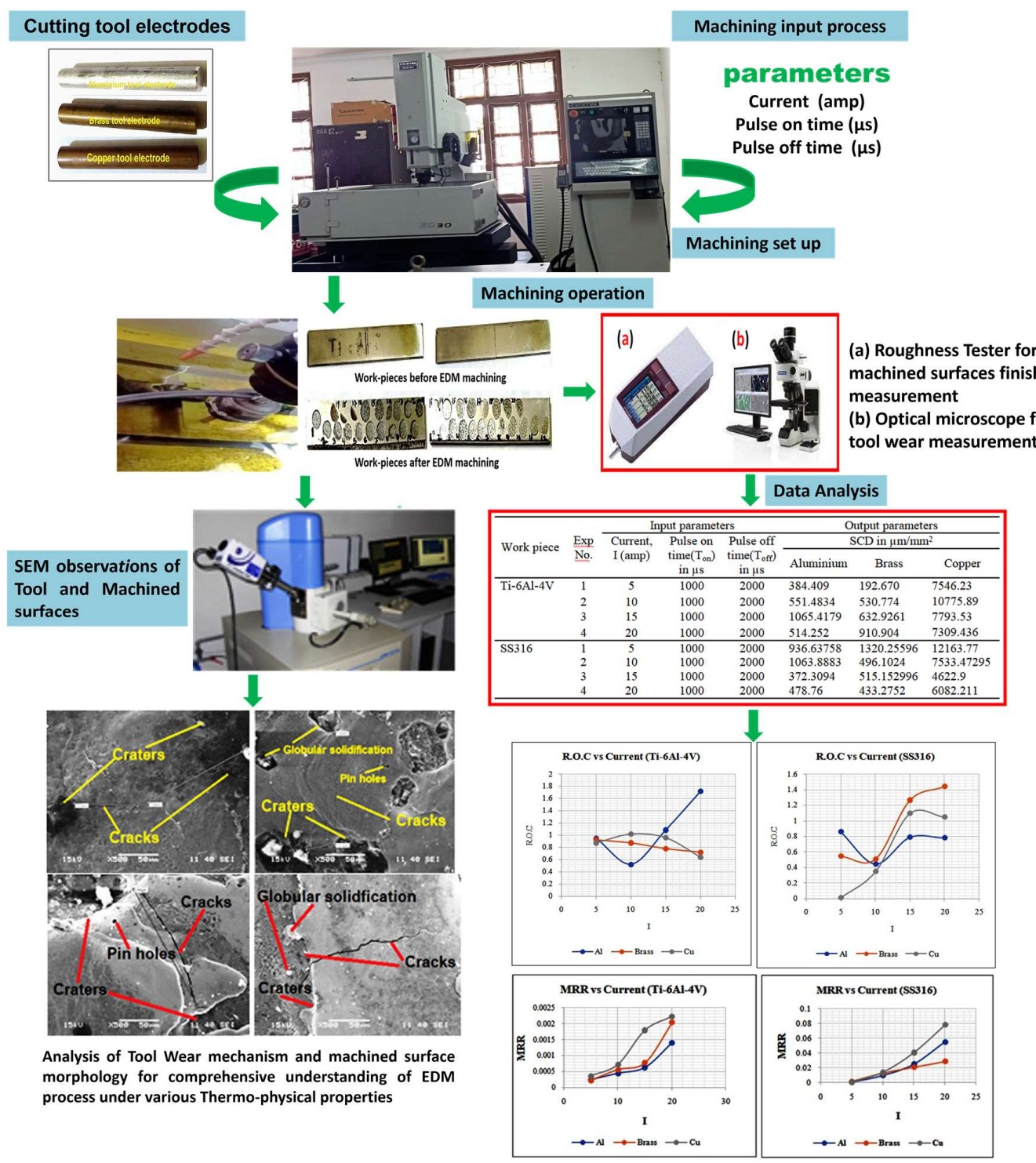

**Figure 3.** Layout of experimental set-up and methodology proposed.

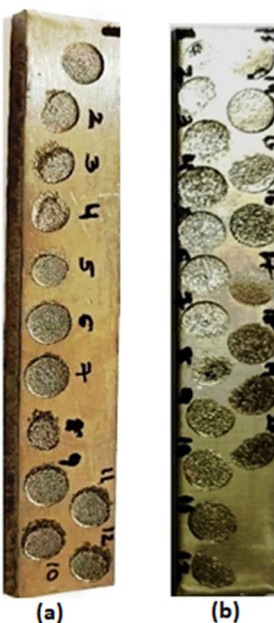

**Figure 4.** Sample work pieces after machining. (**a**) Ti-6Al-4V. (**b**) SS316.

3.    Measurement of SR

After machining was over, the surface roughness of the machined surface was measured using an SR tester (MITUTOYO's SR equipment SJ.210). Only the Ra value is recorded in the results table. It is nothing but the average of the height of the peaks and valleys considered at equal distance within the transverse length of the measuring stylus.

4.    Measurement of ROC

The radial overcut is nothing but the overcut beyond the dimension of the tool on the work piece. Generally, the groove produced on the work piece is of a higher dimension compared to the dimension of the tool tip. The total dimension of the groove was recorded with the help of a coordinate measuring machine (CMM), and the dimension of the tool was recorded using a micrometer. Using these dimensions and applying Equation (3), the radial overcut is determined for each experiment.

$$\text{Radial overcut (ROC)} = \frac{D - d}{2} \text{ mm} \tag{3}$$

where, D = Diameter of the hole and d = Diameter of the tool

5.    Measurement of SCD

After machining was over, the machined surface was viewed under SEM and the total lengths of the cracks were approximately measured in the focusing zone of the microscope with the help of ImageJ software. The area was also recorded, knowing the length and breadth under the focused zone. The following formula is used to determine the SCD.

$$\text{Surface crack density (SCD)} = \frac{l_1 + l_2 + l_3 + \ldots + l_n}{A} \tag{4}$$

where, $l_1, l_2, l_3, \ldots . l_n$ are length of cracks in μm and A is the focused area under the SEM microscope lens in $(\mu m)^2$. This area is calculated by multiplying length and breadth, whose dimensions are shown by SEM microscope.

The different output parameters are calculated by following Equations (1) to (4). At first, the values of MRR considering two work pieces and three different tools are presented in Table 6.

**Table 6.** The values of MRR for different tool–work piece combinations for different experiments.

| Work Piece | Exp No. | Output Parameters | | |
| | | MRR in g/min | | |
| | | Aluminium | Brass | Copper |
|---|---|---|---|---|
| Ti-6Al-4V | 1 | 0.000243 | 0.000219 | 0.00037 |
| | 2 | 0.000446 | 0.000559 | 0.000728 |
| | 3 | 0.000628 | 0.000772 | 0.0018 |
| | 4 | 0.001401 | 0.00204 | 0.00223 |
| SS316 | 1 | 0.000838 | 0.000947 | 0.00027 |
| | 2 | 0.00971 | 0.012436 | 0.0137 |
| | 3 | 0.02491 | 0.020539 | 0.0405 |
| | 4 | 0.05499 | 0.02836 | 0.078 |

The values of tool-wear rate for different work piece–tool combinations for different experiments are presented in Table 7.

**Table 7.** The values of TWR for different tool–work piece combinations for different experiments.

| Work Piece | Exp No. | Output Parameters | | |
| | | TWR in g/min | | |
| | | Aluminium | Brass | Copper |
|---|---|---|---|---|
| Ti-6Al-4V | 1 | 0.000164 | 0.003164 | 0.00002 |
| | 2 | 0.0017 | 0.007044 | 0.00018 |
| | 3 | 0.00197 | 0.011705 | 0.0007 |
| | 4 | 0.0043 | 0.04035 | 0.003 |
| SS316 | 1 | 0.0000125 | 0.001361 | 0.000004264 |
| | 2 | 0.000373 | 0.005589 | 0.00071 |
| | 3 | 0.001056 | 0.010792 | 0.00578 |
| | 4 | 0.00173 | 0.0137 | 0.00454 |

The values of SR for different tool–work piece combinations for different experiments are presented in Table 8.

**Table 8.** The values of SR for different tool–work piece combinations for different experiments.

| Work Piece | Exp No. | SR (*Ra*) in μm | | |
| | | Aluminium | Brass | Copper |
|---|---|---|---|---|
| Ti-6Al-4V | 1 | 2.167 | 2.423 | 3.82 |
| | 2 | 2.3256 | 6.373 | 4.851 |
| | 3 | 2.6545 | 6.942 | 5.697 |
| | 4 | 2.798 | 8.451 | 6.562 |
| SS316 | 1 | 1.51 | 1.662 | 2.376 |
| | 2 | 4.338 | 2.423 | 3.159 |
| | 3 | 7.132 | 2.876 | 6.068 |
| | 4 | 7.209 | 4.1495 | 7.309 |

The values of ROC for different tool–work piece combinations for different experiments are presented in Table 9.

**Table 9.** The values of ROC for different tool–work piece combinations for different experiments.

| Work Piece | Exp No. | ROC in mm | | |
| --- | --- | --- | --- | --- |
| | | Aluminium | Brass | Copper |
| Ti-6Al-4V | 1 | 0.9448 | 0.9239 | 0.8749 |
| | 2 | 0.5239 | 0.8734 | 1.019 |
| | 3 | 1.0857 | 0.7788 | 0.9583 |
| | 4 | 1.7149 | 0.7205 | 0.6409 |
| SS316 | 1 | 0.8589 | 0.5465 | 0.013 |
| | 2 | 0.4465 | 0.5077 | 0.3795 |
| | 3 | 0.7893 | 1.2655 | 1.0975 |
| | 4 | 0.7799 | 1.4413 | 1.0507 |

The values of Vickers micro-hardness (HV) for different work pieces with different tool–work piece combinations are presented in Table 10.

**Table 10.** The values of micro-hardness (HV) for different work pieces with different tool–work piece combinations for different experiments.

| Work Piece | Exp No. | Output Parameters | | |
| --- | --- | --- | --- | --- |
| | | Micro-Hardness in HV | | |
| | | Aluminium | Brass | Copper |
| Ti-6Al-4V | 1 | 1780.7 | 1142.6 | 1777.3 |
| | 2 | 2024.9 | 1837.9 | 2052.7 |
| | 3 | 2276.8 | 1326.3 | 1832.4 |
| | 4 | 1929.0 | 1167.14 | 1667.8 |
| SS316 | 1 | 1165.3 | 1103.3 | 883.9 |
| | 2 | 1204.2 | 1010.1 | 1125.7 |
| | 3 | 1266.7 | 1084.8 | 824.2 |
| | 4 | 1212.9 | 873.0 | 1047.1 |

The values of SCD for different work pieces with different tool–work piece combinations for different experiments are presented in Table 11.

**Table 11.** The values of SCD for different work pieces with different tool–work piece combinations for different experiments.

| Work Piece | Exp No. | Output Parameters | | |
| --- | --- | --- | --- | --- |
| | | SCD in $\mu m/\mu m^2$ | | |
| | | Aluminium | Brass | Copper |
| Ti-6Al-4V | 1 | 384.409 | 192.670 | 7546.23 |
| | 2 | 551.4834 | 530.774 | 10775.89 |
| | 3 | 1065.4179 | 632.9261 | 7793.53 |
| | 4 | 514.252 | 910.904 | 7309.436 |
| SS316 | 1 | 936.63758 | 1320.25596 | 12163.77 |
| | 2 | 1063.8883 | 496.1024 | 7533.47295 |
| | 3 | 372.3094 | 515.152996 | 4622.9 |
| | 4 | 478.76 | 433.2752 | 6082.211 |

## 3. Analysis and Discussion of Experimental Results

Prior to analysis of experimental data, some discussion of SEM photographs was presented as follows:

The SEM photographs have been taken for two work piece materials (i.e., Ti-6Al-4V and SS316 materials) at two different current values (i.e., 5 A and 10 A) by varying the tool

materials as shown in Figures 5a–f and 6a–f. The effects of the tool materials on the surface morphology of the work piece materials have been studied. It is observed that for both the work piece materials there is huge amount of crack density when the Cu tool is used. This is because Cu has high electrical conductivity, which allows higher current to flow through it, resulting in higher MRR, higher thermal stress and hence higher surface-crack density. A better surface morphology is observed in case of Al as compared to Cu and brass. In view of this, it may be recommended that the work piece should be machined first with Cu and finally be machined with Al to obtain higher MRR with better surface morphology.

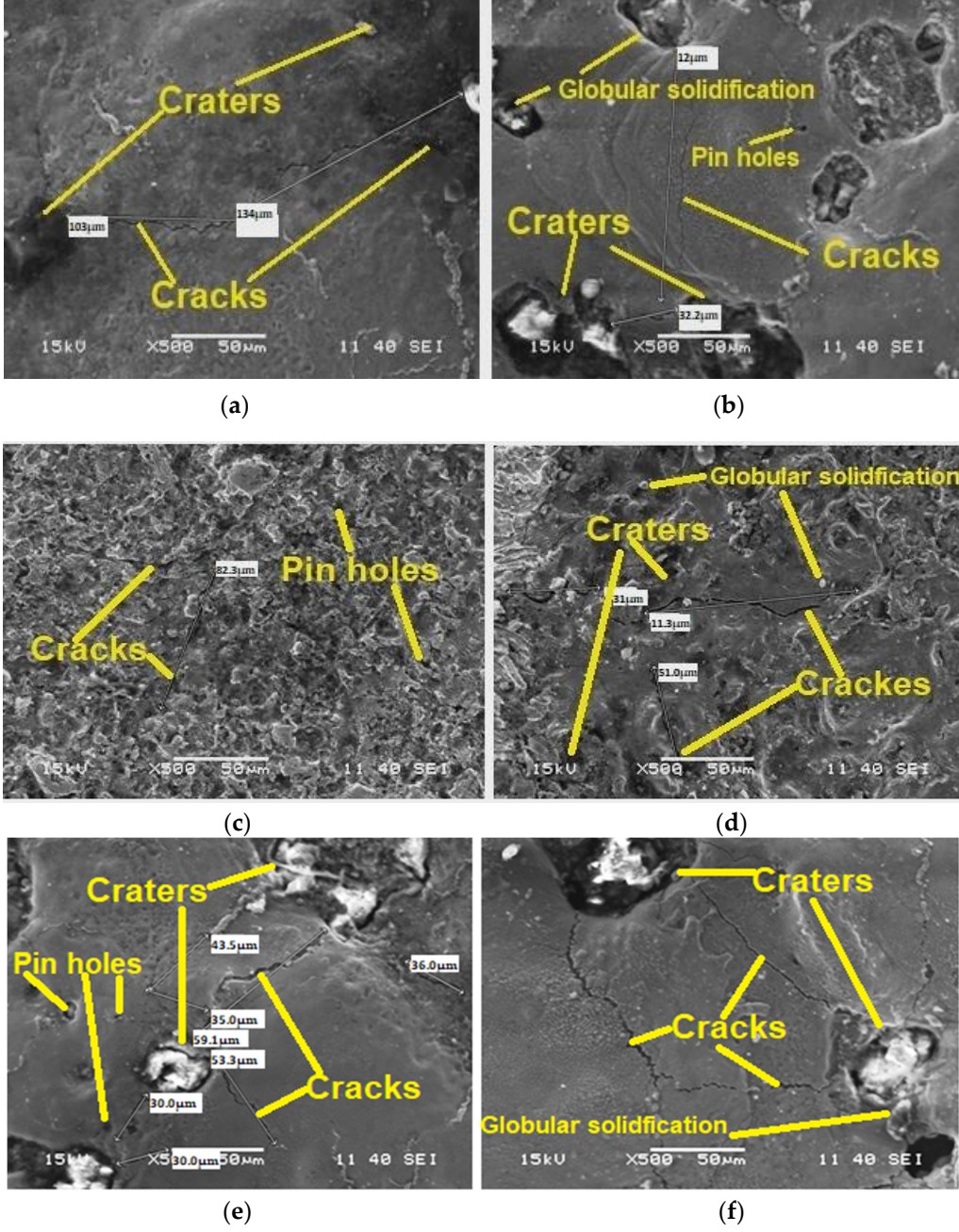

**Figure 5.** SEM micrograph of Ti-6Al-4V machined by Al, Brass and Cu electrode with 500× magnification. (**a**) Al tool tip with I = 5 A, $T_{on}$ = 1000 μs, $T_{off}$ = 2000 μs. (**b**) Al tool tip with I = 10 A, $T_{on}$ = 1000 μs, $T_{off}$ = 2000 μs. (**c**) Brass tool tip with I = 5 A, $T_{on}$ = 1000 μs, $T_{off}$ = 2000 μs. (**d**) Brass tool tip with I = 10 A, $T_{on}$ = 1000 μs, $T_{off}$ = 2000 μs. (**e**) Cu tool tip with I = 5 A, $T_{on}$ = 1000 μs, $T_{off}$ = 2000 μs. (**f**) Cu tool tip with I = 10 A, $T_{on}$ = 1000 μs, $T_{off}$ = 2000 μs.

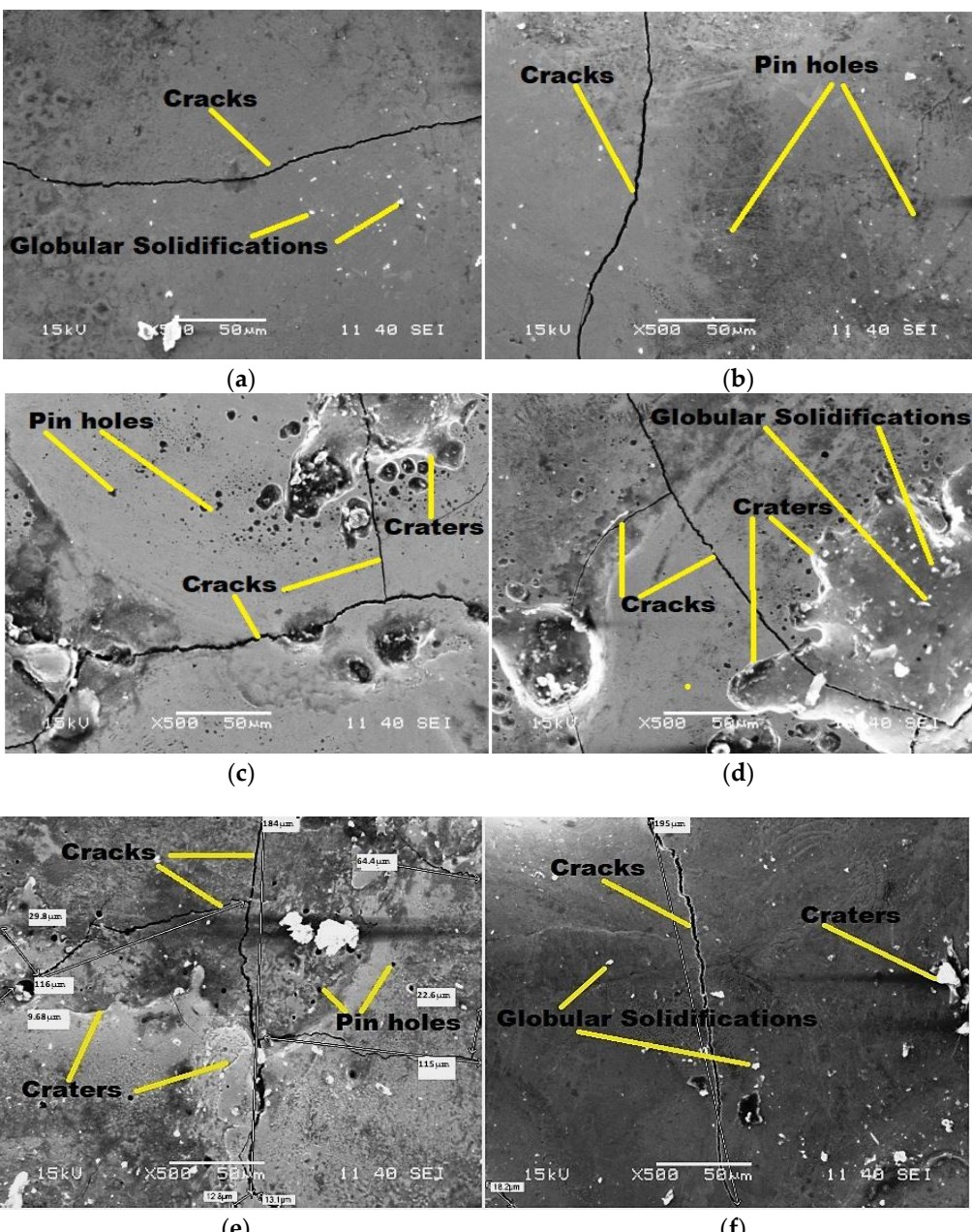

**Figure 6.** SEM micrograph of SS316 machined by Al, Brass and Cu electrode with $500\times$ magnification. (**a**) Cu tool tip with I = 5 A, $T_{on}$ = 1000 μs, $T_{off}$ = 2000 μs. (**b**) Cu tool tip with I = 10 A, $T_{on}$ = 1000 μs, $T_{off}$ = 2000 μs. (**c**) Al tool tip with I = 5 A, $T_{on}$ = 1000 μs, $T_{off}$ = 2000 μs. (**d**) Al tool tip with I = 10 A, $T_{on}$ = 1000 μs, $T_{off}$ = 2000 μs. (**e**) Brass tool tip with I = 5 A, $T_{on}$ = 1000 μs, $T_{off}$ = 2000 μs. (**f**) Brass tool tip with I = 10 A, $T_{on}$ = 1000 μs, $T_{off}$ = 2000 μs.

Furthermore, from Figure 5a,b with an increase in current, the surface finish of the brass electrode will be less. Therefore, the machined surfaces corresponding to these two electrodes (Cu, brass), analyzed under a microscope, are almost identical. Approximate impressions shown in Figure 5c–f confirm the argument, since the dimensions of the craters are almost identical.

From the above, it can be concluded that the Cu electrode is the best of the three electrodes to achieve the maximum erosion rate when machining Ti-6Al-4V. The Cu tool electrode has higher SCD than the brass and aluminum electrodes. This is due to more electrical conductivity of the Cu electrode than the other two electrodes.

In order to determine the overall effect of the tool–work piece combination on the machining performances, the sum of the values of output parameters obtained from four different combination of input parameters were added and presented in Table 12.

**Table 12.** Sum of the values of output parameters.

| Sl No. | Work Piece | Type of Tool | Sum of MRR (g/min) | Sum of TWR (g/min) | Sum of SR (µm) | Sum of ROC (mm) |
|---|---|---|---|---|---|---|
| 1 |  | Aluminium | 0.002718 | 0.008134 | 10.4835 | 4.2693 |
| 2 | Ti-6Al-4V | Brass | 0.00359 | 0.062263 | 24.189 | 3.2966 |
| 3 |  | Copper | 0.005128 | 0.0039 | 20.93 | 3.4931 |
| 4 |  | Aluminium | 0.090448 | 0.0031715 | 20.189 | 2.8746 |
| 5 | SS316 | Brass | 0.062282 | 0.031442 | 11.1105 | 3.761 |
| 6 |  | Copper | 0.13247 | 0.011034264 | 18.912 | 2.5092 |

It is observed from Table 12 that, for machining Ti-6Al-4V, Cu showed the best performance compared to brass and Al. The Cu electrode had the highest MRR, lowest TWR, intermediate SR and intermediate ROC. Al showed the worst performance, whereas the performance of brass is in between Cu and Al. It is observed that the Cu tool material had the highest electrical conductivity, thermal conductivity and thermal diffusivity, and highest melting point, showing best performance.

Similarly, for machining SS316, the same Cu tool material showed the best performance compared to Al and brass. The Cu electrode had the highest MRR, intermediate TWR, intermediate SR and lowest ROC. Brass showed the worst performance because of the lowest MRR, highest TWR, lowest SR and highest ROC. However, Al showed intermediate performance because of the intermediate MRR, lowest TWR, highest SR and highest ROC. The best performance is Cu due to the same reason stated for the Ti-6Al-4V material.

For displaying the effects of different tool materials on MRR, TWR, SR, ROC, SH and SCD for two different work piece materials, different sets of graphs have been plotted, as shown in Figures 7–12. In each case, the variations of performance characteristics have been presented with variation of current, considering different tools.

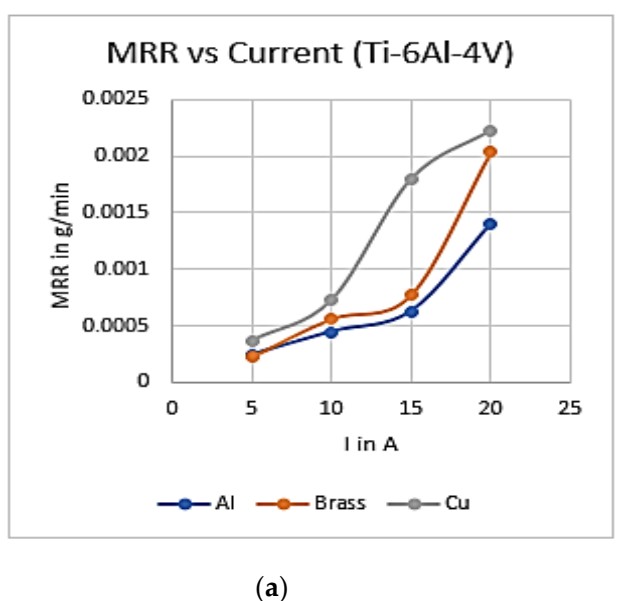

(**a**)

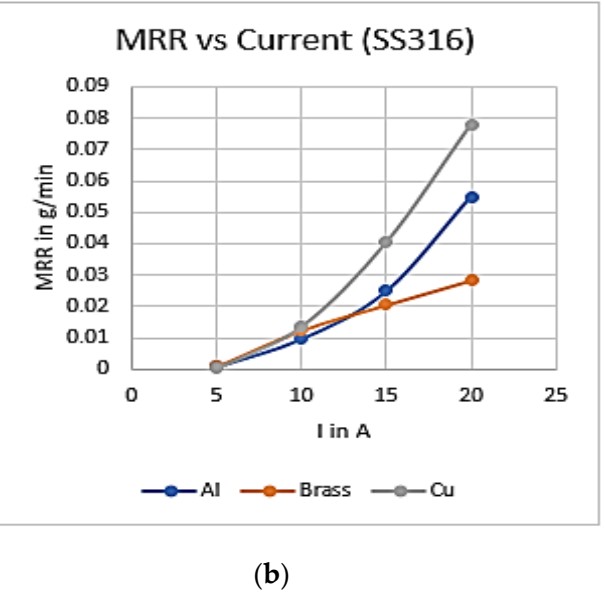

(**b**)

**Figure 7.** Variations of MRR for (**a**) Ti-6Al-4V and (**b**) SS316 using different tools.

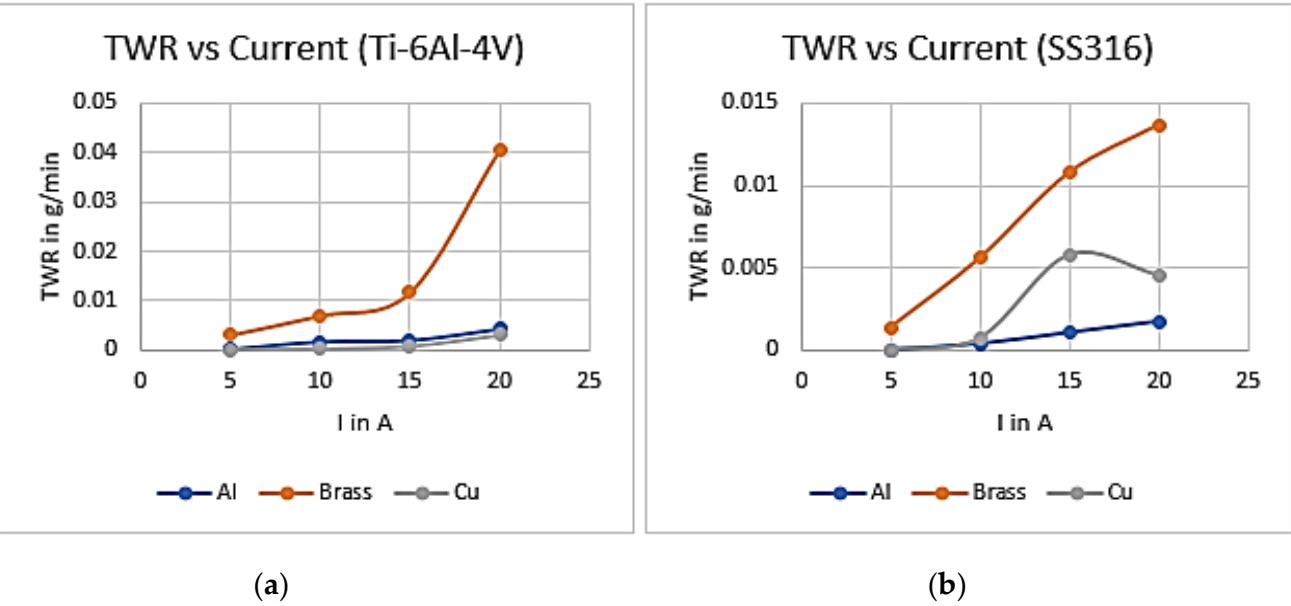

**Figure 8.** Variations of TWR for (**a**) Ti-6Al-4V and (**b**) SS316 using different tools.

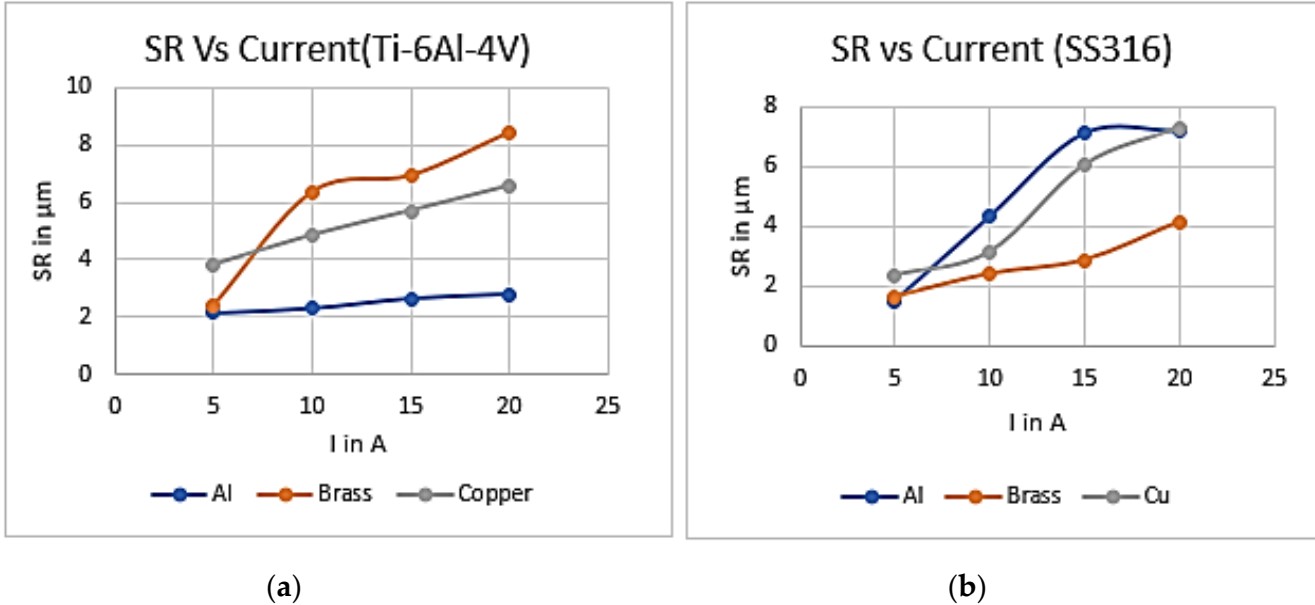

**Figure 9.** Variations of SR for (**a**) Ti-6Al-4V and (**b**) SS316 using different tools.

Figure 7a represents the variations of MRR of Ti-6Al-4V material at different currents using different tool materials. It is clearly obvious from this graph that the MRR values drastically increase with current when the Cu tool is used compared to the Al and brass tools. Hence, Cu is the best tool material among these three tools so far as the MRR is concerned. The reason for higher MRR in the case of Cu is explained earlier.

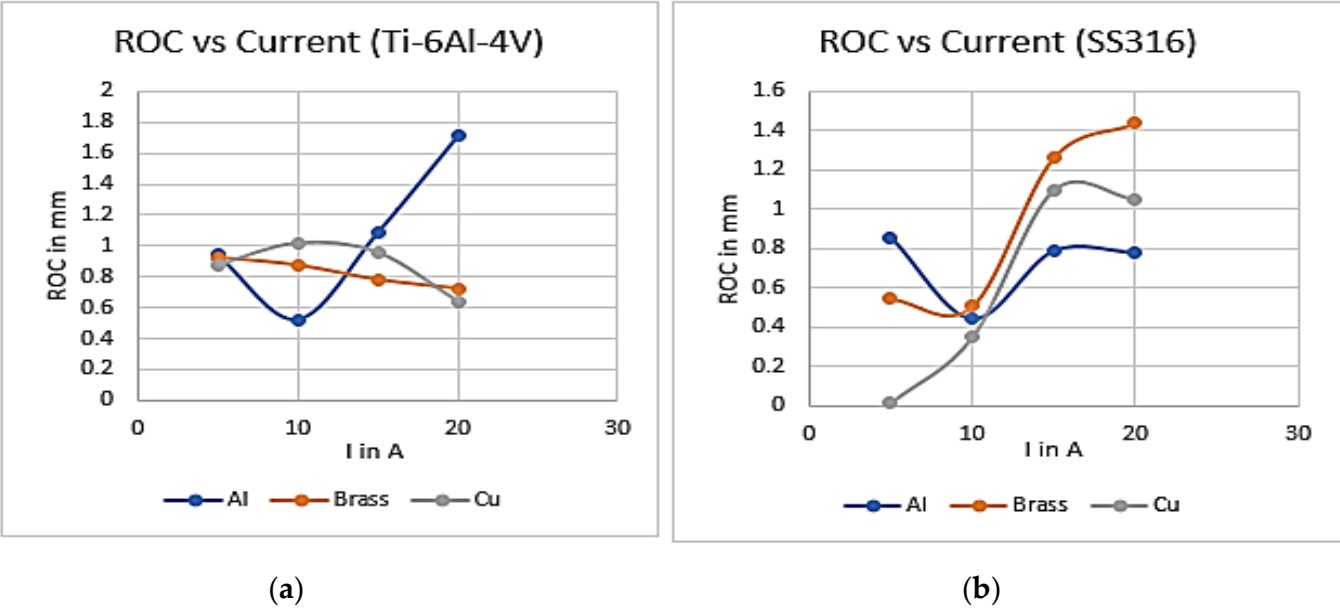

**Figure 10.** Variations of ROC for (**a**) Ti-6Al-4V and (**b**) SS316 using different tools.

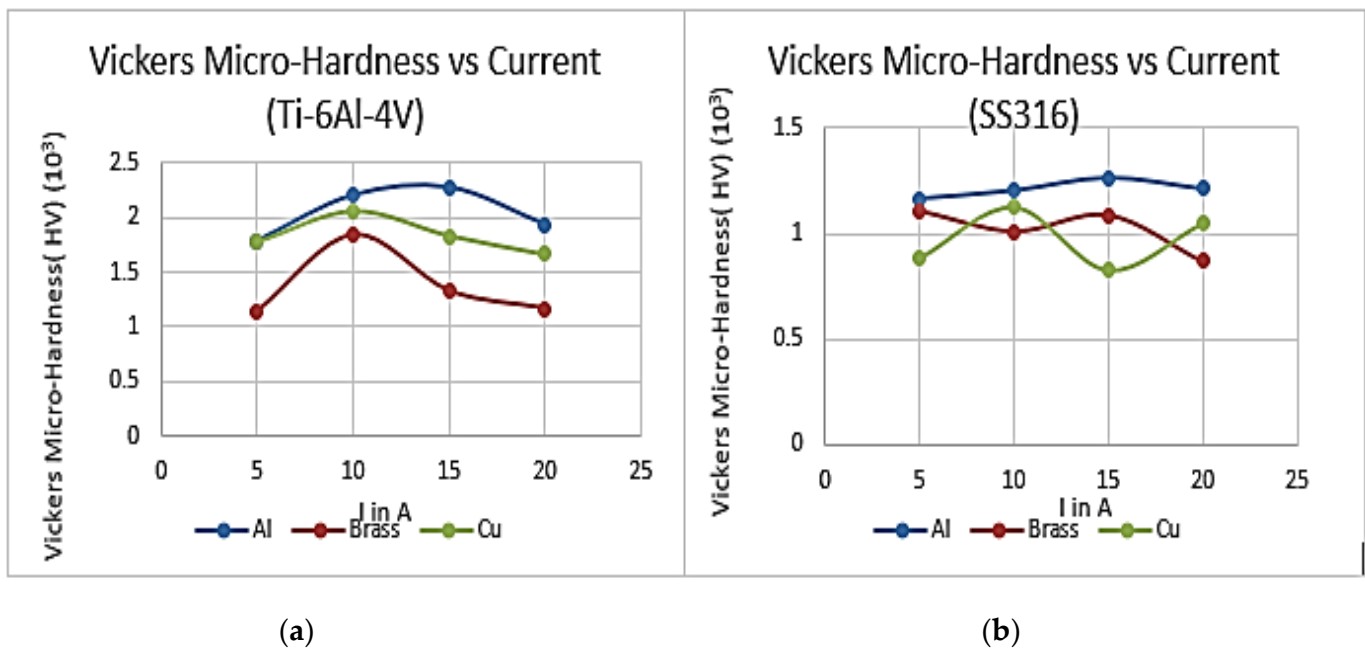

**Figure 11.** Variations of VH for (**a**) *Ti-6Al-4V* and (**b**) *SS*316 using different tools.

Figure 7b represents the variations of MRR of SS316 material at different currents using different tool materials. It is shown in the above graph that the MRR values drastically increase with current when the Cu tool is used compared to Al and brass tools. Hence, Cu is a best tool material among these three tools with regards to MRR of the work piece material. The reason is same as for Ti-6Al-4V.

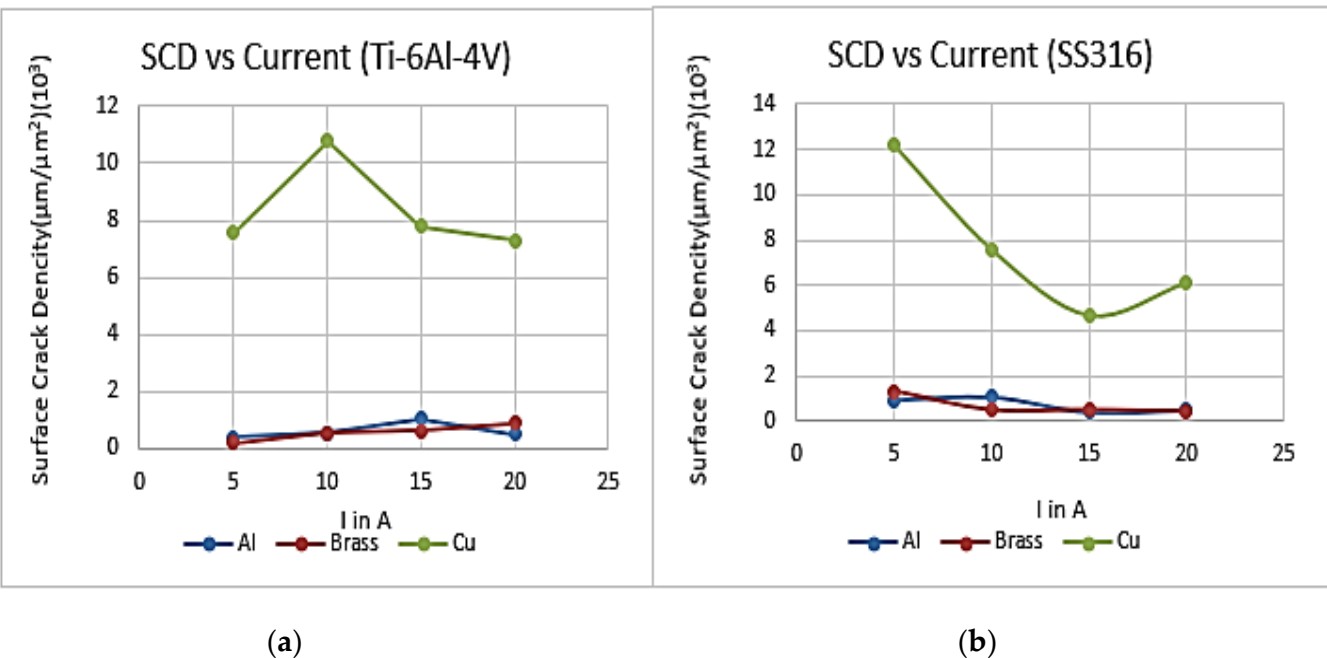

**Figure 12.** Variations of SCD for (**a**) Ti-6Al-4V and (**b**) SS316 using different tools.

It is observed from Figure 8a that when Al tool material is used, the TWR at low current is very high as compared to Cu and brass. However, when the current increases beyond 15A, the TWR drastically increases. The value of TWR is very low for brass and Cu and also increases very slowly with an increase in current. In comparison to brass, Cu is exhibiting less TWR and hence Cu may be considered as the best tool material for machining Ti-6Al-4V material among these three tools, when TWR is taken into consideration. It may be due to the highest melting point as well as the highest thermal conductivity value.

The variation of TWR with respect to current while machining SS316 is represented in Figure 8b. Among the three tool materials, Cu is exhibiting minimum TWR and the increase rate of TWR with respect to current (I) is also very low. It is also due to its higher thermal conductivity value and highest melting point value. Hence, Cu may be treated as the better tool material among these tools with respect to TWR. The reason is same as stated for the Ti-6Al-4V material.

Figure 9a represents the variations of SR with an increase in current when three different tool materials such as Al, brass and Cu were considered for machining the Ti-6Al-4V material. It is seen that Al shows good surface finish as compared to brass and Cu. The SR of Cu is intermediate between Al and brass. The brass is showing the highest SR value of the three materials. This happened due to the minimum values of thermal conductivity and diffusivity. Figure 9b represents the variations of SR with respect to current when three different tool materials such as Al, brass and Cu are used for machining SS316. In this figure it is clearly observed that Al shows the highest SR value compared to Cu and brass. The brass shows better surface finish compared to the other two materials. The reason is the same as stated above.

Figure 10a represents the variations of the ROC with increases in current when the Ti-6Al-4V sample is machined with three different tool materials. The variation of ROC for Cu and brass is more or less the same and it is decreasing with increasing current. However, when Al is used as a tool material, the ROC value initially decreases from a 5A to a 10 A current, and subsequently increases rapidly when the value of the current increases. At a 10A current, Al shows the minimum ROC value compared to the other tool materials. The decrease in the ROC value for both brass and copper at a higher current (20A) with respect to lower current (10A) may be due to the pinch effect or less spreading with high current.

Figure 10b represents the variations of the ROC values with increase in current when the SS316 sample is machined with the help of Al, brass and Cu. The nature of variation is more or less the same for the three different tool materials. It is observed that when the current value is low, Cu is the best tool material with respect to ROC. Considering all three tool materials, the overall average value of ROC in the case of Ti-6Al-4V is more compared to SS316. This is due to the higher thermal diffusivity value of the Ti-6Al-4V material as compared to SS316. The ROC corresponding much less to the 10A current may be due to the higher pinch effect.

The variation of Vickers micro-hardness value (HV) with variation of current (I) are presented in Figure 11a,b for Ti-6Al-4V and SS316 sample materials, respectively. It is observed that, while machining Ti-6Al-4V with Al, brass and Cu tool materials, the HV value first increases with an increase in I to a peak value, and then decreases. The peak value for Al is the highest. The optimal value of I for achieving higher SH is 10 to 12 A. This may be due to the following reason. At a low value of I, the tool material is less heated, and as a result, a lower amount of phase transformation takes place and hence the low value of hardness. Similarly, at a higher value of I, though the phase transformation takes place due to a lesser quenching effect, the hardness is also decreased. Hence, the optimum current is to be maintained for obtaining higher hardness of the tool. With reference to Figure 11a, it appears that for machining SS316 with brass and Cu, there is more fluctuation in the micro-hardness values of these tool materials. However, when Al is used, the micro-hardness values initially increase marginally and decrease slowly with increase in current. It is noticed from the average micro-hardness values (as shown in Table 9) for both the work piece materials, considering three different tool materials, aluminium shows the highest average value. Although Al and Cu have more or less the same reflectivity value, Al has a lower thermal conductivity value compared to Cu, as a result of which the amount of heat flux remaining on the work piece when Al is used as the tool is more compared to Cu. This will result in more phase transformation and will ultimately develop a more-hardened surface after cooling.

$T_{off}$ and current were found to be the most significant parameters influencing SR and SH, respectively [8], whereas $T_{off}$ and current significantly influenced MRR, as discussed by Chaudhari et al. [30]. Sanchez et al. found computer software for the simulation of wire deformation in wire EDM in taper-cutting of hard materials [31]. Sanchez et al. had examined that if the higher levels of accuracy are required, a strategy based on cutting regime modification combined with finishing cuts must be used [32].

The variations of the SCD with an increase in I for two work piece materials, that is, Ti-6Al-4V and SS316, are illustrated in Figure 12a,b, respectively. When both the work pieces are machined with the Cu electrode, there is a wide variation of SCD with an increase in current. This may be due to its highest thermal conductivity value. Since it is desirable to have a minimum SCD, the appropriate I for both the work piece materials while machining with Cu is about 15A. The SCD for Al and brass materials is very low for machining Ti-6Al-4V and SS316 materials.

## 4. Conclusions

The following conclusions are obtained from the present work:

- Among Al, brass and Cu, the copper tool has the highest melting point, highest thermal conductivity, highest thermal diffusivity and highest electrical conductivity, as a result of which it shows the best machining performance for Ti-6Al-4V materials due to the highest MRR, lowest tool-wear rate, intermediate ROC and SR values.
- Similarly, Cu also shows the best machining performance for machining SS316 due to the highest MRR, lowest TWR, intermediate surface roughness and ROC values.
- Due to the lowest thermal conductivity and diffusivity of brass among Al, brass and Cu, it shows the worst machining performance. Therefore, it is not recommended for machining Ti-6Al-4V and SS316 work piece materials.

- A current of 15 A is the optimum value for achieving the highest micro-hardness value and lowest surface-crack density for these three tool materials.
- Although the Cu tool is better with respect to MRR and TWR, it shows the highest SCD while machining Ti-6Al-4V. This may be due to the good electrical conductivity of Cu, which allows the maximum possible current and hence higher thermal stress on Ti-6Al-4V due to the lower melting point value of Ti-6Al-4V as compared to the SS316 material.

**Author Contributions:** S.S., writing—original draft preparation, data curation; R.K.B., methodology, Conceptualization, writing—review and editing; J.P.D., formal analysis, supervision; J.R., validation, writing—review and editing. All authors have read and agreed to the published version of the manuscript.

**Funding:** This research received no external funding.

**Conflicts of Interest:** The authors declare no conflict of interest.

## Nomenclature

| | |
|---|---|
| A | Ampere |
| D | Diameter of the hole |
| d | Diameter of the tool |
| EDS | Energy Dispersive Spectroscopy |
| EDM | Electro-discharge machining |
| HV | Micro-hardness Value |
| I | Current (Ampere) |
| $l_1$, $l_2$, $l_3$ | Length of crack surface |
| M. P. | Melting point |
| MRR | Material Removal Rate (g/min) |
| ROC | Radial overcut (mm) |
| SR | Surface roughness (μm) |
| SCD | Surface crack density μm/(μm)$^2$ |
| SH | Surface Hardness |
| SEM | Scanning electron microscope |
| TPP | Thermo-physical properties |
| TWR | Tool wear rate (g/min) |
| $T_{on}$ | Pulse on time (μs) |
| $T_{off}$ | Pulse off time (μs) |
| Wt. | Weight |

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
