# Peer review of "Effect of Thermo-Physical Properties of the Tool Materials on the Electro-Discharge Machining Performance of Ti-6Al-4V and SS316 Work Piece Materials"

_jmmp, doi:10.3390/jmmp6050096_

Round 1

Reviewer 1 Report (Previous Reviewer 1)

The authors have improved the article with previously suggested comments and can now be published.

Reviewer 2 Report (Previous Reviewer 2)

Ok

This manuscript is a resubmission of an earlier submission. The following is a list of the peer review reports and author responses from that submission.

Round 1

Reviewer 1 Report

The reviewer comments of the paper

«Individual tool materials used in EDM process by changing the thermo-physical properties for evaluation of the machining performance»

- Reviewer

The authors presented an article «Individual tool materials used in EDM process by changing the thermo-physical properties for evaluation of the machining performance». However, there are several points in the article that require further explanation.

Comment 1:

The abstract needs to be improved.

Demonstrate in the abstract novelty, practical significance. Add quantitative and qualitative work results to the abstract. Describe briefly in the abstract what kind of alloys are they SS316 and Ti-6Al-4V?

Comment 2:

The introduction needs to be improved.

Firstly, group quotation is unacceptable in one phrase, for example [28-30], [31-34]. Break this sentence into parts or individual sentences. For example, ... [...], ... [...], etc. Or one reference - one sentence.

Now the list of references needs to be supplemented with at least 6-8 more references published over the past 5 years. Here are some recent articles:

Metals 2021, 11(11), 1668. doi:10.3390/met11111668

Journal of Manufacturing Processes 2021, 64, 1105-1142. doi:10.1016/j.jmapro.2021.01.056

Materials 2021, 14(21), 6420. doi:10.3390/ma14216420

After analyzing the literature, show before formulating the goal of the "blank" spots. Which has not been previously done by other researchers. You must show the importance of the research being undertaken. Show what will be the new research approach in this article. You need to show a hypothesis.

Add a clear purpose to the article.

Comment 3:

2. Materials and Methods

Are all figures original? If not needed appropriate citations and permissions. Refine this for figures throughout the article.

Are all formulas original? If not needed appropriate citations.

What is the hardness of the workpiece and how was it measured?

The quality and resolution of all figures needs to be improved. Now they are vague and not clear to the reader.

Describe the measurement procedure in more detail. At what point in time? How is the measuring setup set up? How many repetitions of measurements? What statistical methods are used to process experimental results? Describe the experimental stand in more detail. What method of experiment planning is used and why?

Comment 4:

3. Results and discussion

Add a legend for each table 5,6,7,8,9,10. At least two or three sentences for each table.

Reference to figure 6 was not found in the text of the article.

The table or figure should appear immediately after the paragraph where it was mentioned. Pay attention to table 11.

What is the difference from previous work in this area?

Comment 5:

It will be useful to add a section of Nomenclature in which to sign all the physical quantities and abbreviations encountered in the article. There are many physical quantities in the text and such a section will help to find the description of the necessary element.

For example,

p                : Density (g/cm3)

MRR         : Materials removal rate

etc.

Ra (surface roughness). Give the definition in accordance with ISO. At least in one nomenclature.

Comment 6:

Conclusions needs to be improved.

It is necessary to more clearly show the novelty of the article and the advantages of the proposed method. Show practical relevance

The article is interesting, but needs to be improved. Authors should carefully study the comments and make improvements to the article step by step. After major changes can an article be considered for publication in the "Journal of Manufacturing and Materials Processing".

Reviewer 2 Report

You need to rework paper carefully, please read the attached file and follow all the suggestions.

Reviewer 3 Report

Comment for ID# jmmp-1824345:

1. The topic concerns three tool materials (by changing their thermo-physical properties?) for evaluating the EDM machining performance.

 1-1. But, it is strange enough to compare copper, and brass with aluminum, which is rare to use as a tool electrode in the EDM fields with relatively low melting point (about 31.2% K less than that of copper). There seems lack of explanation on the reason for choosing Al as a comparative tool electrode.

 1-2. The title of " individual tool materials ...by changing their thermo-physical properties" is mis-leading the readers that the tool materials adopted in this study are not subjected to changing their thermo-physical properties, but with their different thermo-physical properties, as stated in Table 3.

2. As stated in the paragraph just after Table 3 :

"Since Copper has the highest melting point with very high thermal conductivity and diffusivity as compared to brass and aluminum, its wear rate will be less and durability will be more."

It looks like trivial knowledge or even a comment sense to the EDM manufacturing industries, that coper is better than brass and that Al is never considered as a tool, which further weakens the meaning and the importance of this research.

3. In Table 4, the Input parameter sets include current (peak) I, Ton, and Toff. The parameters Ton is fixed at 1 ms and Toff is fixed at 2 ms, with varying current from 5 to 20 A. However, the EDM machining performance includes as many as six indexes, as stated in Line 174 of Page5. That simplifies the procedures is observing the effect of EDM current's contribution on the six performance indexes.

The relevant EDM operation conditions to the performance indexes, such as SCD, SH, TWR, and ROC, not only depend on the Input parameter sets of current I, Ton, and Toff, but also the mechanical and electrical conditions. For example, the dielectric flushing condition, tool jumping parameters, and servo feeding methods affect the machining performance significantly. Nevertheless, there is no role played in this report without explanation.

4. The SCD equation of (ii) is ambiguous that why there are typically three crack lengths l1, l2 to l3, and no definition of the  "Area".

5. The MRR equation of (iv) is unreadable; that What is Fi ____ wt. of workpiece in Line 205, and the statement followed by this equation needs to be rewritten by proper definition.

6. Results:

6-1. please make a statement before the related Figures for each paragraph.

6-2. In Fig.11, the SR reflects the surface roughness. But SR with Al tool is unlikely reasonable, especially. Why is that 15A current can result in a a less SR than 10A?

6-3. In Fig.13, the ROC reflects the radial over-cut. The trend for Brass tools over 10A and the trend for copper tools over 15A are also reasonable in that higher peak current results in less over cutting radius (20A may create almost twice the energy of 10A EDM). Or maybe there are some unmodeled factors affecting these results. 

6-4. Similarly, In Fig.14, the ROC response of 316 steel is very interesting at peak current 10A. However, why is that?

6-5. The inference of Section 3-1. Grey Relational Analysis may not be feasible because the input sets only vary the current parameter that is almost impossible to get enough testing for the purpose of optimization from the Grey Relational Analysis. 

6-6. In Fig.21 - Fig.22, Why the significance threshold was set at 0.005, which is set to a very low level that is not the usual threshold. To check the statistics, how about Pearson’s relative coefficient or Spearman’s rho?

7-1. The conclusion points 1-2 seem very trivial to the EDM manufacturing industries and academic fields.

7-2. The conclusion point 4, the p-value for the TWR testing is as low as 0.014, revealing a very little normal-probability relation.

7-2. The conclusion point 5, the comparison meaning is unclear since the definition of SCD stated in equation (ii) is not adequately defined yet.

8. Reference survey:

8-1. The format of the Reference list should be carefully corrected and adjusted to a consistent format. Ex. year and volume of the reference [3].

8-2. Line 49 on Page2, [3] is miss-placed, that two materials are adopted as workpiece in this study, but not in paper [3].

8-3. Line 51 on Page2, paper [4] is irrelevant to the topics of this study. That erosion test is not correlated to the mentioned EDM process and tool materials.